# Putting cumulative (dis)advantages in context: Comparing the role of educational inequality in later-life functional health trajectories in England and Germany

Martin Wetzel[1]*, Bram Vanhoutte[2,3]

**1** University of Cologne, Cologne, Germany, **2** École de Santé Publique, Université Libre de Bruxelles, Brussels, Belgium, **3** Department of Sociology, University of Manchester, Manchester, United Kingdom

* wetzel@wiso.uni-koeln.de

## Abstract

### Background

The cumulative (dis)advantage (CAD) perspective more and more is examined in a comparative way, to highlight the role of context in generating inequality over the life course. This study adds to this field of research by examining trajectories of activities of daily living (ADL) in later life by educational level in a country comparison of England and Germany, emphasizing differing institutional conditions.

### Method

Data used are the English Longitudinal Study of Ageing (ELSA; 11,352 individuals) and the German subsample of the Survey of Health and Retirement in Europe (SHARE; 5,573 individuals). Using population averaged Poisson panel regression models, 12-year trajectories of six birth cohorts are investigated, covering the age range of 50 to 90 years.

### Results

Respondents in England have a higher level of limitations at age 50, and more limited increases over age than in Germany. An educational gradient exists in both countries at age 50. Notably, the educational gradient increases for more recently born cohorts, but declines with increasing age in England, while in Germany educational differences increase for more recently born cohort only.

### Discussion

The current study indicates that CAD processes between educational groups are context sensitive. While England showed convergence of disparities with increasing age, in Germany no differential development was found.

**Data Availability Statement:** The data cannot be uploaded in a repository due to data protection policies of SHARE and ELSA. We cite from the SHARE website: "Data users are not allowed to

make copies of the data available to others and/or enable any third party access to the database." However, data are available from SHARE Research Data Center (for further instructions see www. share-project.org/data-access/user-registration. html) and UK Data Service (see www. ukdataservice.ac.uk for study ID 8444). In a first step to harmonize both data sets, we applied the code provided by Gateway to Global Ageing project (see www.g2aging.org/?section=downloads for the "Harmonized Stata Code" for SHARE (version C.2) and ELSA (version F)). Further, code necessary to replicate the data management, analyses, and data extraction (including tables and graphs) can be found under https://osf.io/dks4t/?view_only= 89c7a7252b054ad9b5ebda83a444fe25. In this way other researchers can register with SHARE and ELSA for free, download the data set, run our code and reproduce all our results.

**Funding:** Major parts of this work originated during a research stay of Martin Wetzel at the Cathie Marsh Institute of the University of Manchester which was supported by a mobility grant of the Excellence Initiative of University of Cologne. Martin Wetzel is currently working in the pairfam-project which is funded by the German Research Foundation (DFG). Bram Vanhoutte was funded by the fRaill project (grant MRC G1001375/1) as part of the cross research council Life Long Health and Wellbeing programme, and was supported by a Simon Fellowship of the University of Manchester, as well as a Derby Fellowship of the University of Liverpool. The funders had no role in study design, data collection and analysis, decision to publish, or preparation of the manuscript.

**Competing interests:** The authors have declared that no competing interests exist.

# Introduction

"One and the same cause wears out our bodies and our clothes", a worker says to his doctor in a poem by German playwright Bertolt Brecht, aptly describing how social position, as a fundamental cause, affects physical health and its decline. Cumulative (dis)advantage (CAD), or the idea derived from Merton's Matthew effect that advantage leads to opportunity, while disadvantage leads to risk, has often been used to characterise how social inequalities affect health development over time [1, 2]. While the three central tenets to investigating CAD identified by Wilson, Shuey and Elder [3], namely 1) adopting a life course perspective, 2) distinguishing age from cohort effects, and 3) tackling selective panel dropout, still hold, attention has shifted to a new aspect of CAD. Previously, studies testing the CAD hypothesis focused exclusively on one institutional context (e.g., [3–27]). However, a new wave of research [28–34] strengthens the argument that a fourth central principle, namely how country context modifies the way in which CAD processes unfold over time, is needed to further a deeper understanding of CAD as a dynamic, contextualised process within social systems.

To do so, the current study contextualises the relation between educational level and development of functional health limitations over the age of 50 to 90, by focusing on two European countries: England and Germany. Combining two sets of longitudinal data, the English Longitudinal Study of Ageing (ELSA) and the German Subsample of the Survey of Health and Retirement in Europe (SHARE), we examine 12-year functional health trajectories for cohorts born between 1917 and 1965. We intend to show how interpretations of CAD processes profit from a comparative angle, since the extent to which CAD produces inequalities in later life needs context to become meaningful.

# Background: The country context of cumulative advantage and disadvantage (CAD) in health

## Country context

While context is key in much life course research, CAD processes in health up until recently were often examined within one country, usually the US (e.g., [22–24, 26, 29], see Table 1). This is mainly due to extensive data demands, in the form of large, high quality, longitudinal panel data spanning decades, but also the advanced methodological nature of the CAD debate, which often means restricting analysis to a well know example. As a high quality data infrastructure on ageing has matured, and a consensus on how to investigate CAD has emerged, exemplified by Wilson, Shuey and Elder's three central tenets, the occurrence and strength of this process has been tested in contexts other than the US: across the whole of Europe [25, 35], in individual European countries such as Germany [19], Sweden [18] and Switzerland [5], as well as comparing several European countries with contrasting positions in terms of their welfare systems [32, 33]. These studies illustrate that the US context is exceptionally inductive to CAD processes, combining high initial levels of health problems, with a large educational health gap, both widening over time within cohorts, as well as widening up for younger cohorts [33]. In contrast, European countries show a more mixed picture, with an educational health gap that diverge over age, which is less pronounced in more egalitarian countries, and which is not universally wider for younger cohorts [19, 33].

This study wants to expand the research on CAD processes in Europe, by tracing country differences in two countries, England and Germany, but without relying on the often used worlds of welfare narrative [45]. While countries like Sweden and the US might fit their welfare regime ideal types well, clearly belonging to government oriented social democratic and market-oriented liberal welfare states respectively [46], these typologies tend to oversimplify

**Table 1. Overview over studies on health (inequality) since 2007.**

| Author(s), Year | | Health dimension | Development | Cohort | Panel attrition | SES | Race | Sex | Early Life | Country |
|---|---|---|---|---|---|---|---|---|---|---|
| *Health-development studies (selection)* | | | | | | | | | | |
| Spence et al., 2011 | [36] | M | X | | X | X | X | X | | US |
| Haas & Oi, 2018 | [37] | F, P | X | X | | X | | X | X | 13# |
| Lin, 2020 | [38] | F | X | | X | X | X | X | | US |
| *Cohort-sensitive health-development studies (selection)* | | | | | | | | | | |
| Brault, Meuleman, & Bracke, 2012 | [39] | M | X | X | X | | | | | BEL |
| Bell, 2014 | [40] | M | X | X | | X | X | X | | UK |
| Haas, Oi, & Zhout, 2017 | [41] | F | X | X | | | | | | 14# |
| Brailean et al., 2018 | [42] | C | X | X | X | | | | | NL |
| Zaninotto et al., 2018 | [43] | C | X | X | X | | | X | | UK |
| *Cross-sectional CAD studies (selection)* | | | | | | | | | | |
| Schöllgen, Huxhold, & Tesch-Römer, 2010 | [4] | F, P, S | | | | X | | X | | GER |
| Schaan, 2014 | [15] | M | | | | X | | | X | EU |
| Shrira & Litwin, 2014 | [21] | F, M | | | | | | | X | EU |
| Vanhoutte & Nazroo, 2015 | [28] | M | | X | | X | | | X | UK US |
| *Longitudinal CAD studies (selection)* | | | | | | | | | | |
| Mirowsky & Ross, 2008 | [22] | S | X | | X | X | | | | US |
| Haas & Rohlfsen, 2010 | [23] | F | X | | X | X | X | | X | US |
| Liang et al., 2010 | [24] | S | X | | X | | X | | | US |
| McDonough, Worth, & Sacker, 2010 | [29] | S | X | | | X | | | | UK US |
| Leopold & Engelhartdt, 2013 | [25] | F, P, S | X | | | X | | | | EU |
| Pais, 2014 | [26] | o. | X | | X | | X | | X | US |
| Jackson, 2015 | [27] | P, M | X | | X | X | | | X | UK |
| Cullati, 2015 | [44] | F, S, M | X | | | X | | X | | CH |
| McDonough et al., 2015 | [30] | S, M | X | | X | X | | X | X | UK US |
| Wu et al., 2020 | [31] | P | X | | | X | | X | | 8# |
| *Cohort-sensitive CAD studies* | | | | | | | | | | |
| Willson et al., 2007 | [3] | S | X | X | X | X | | X | | US |
| Yang, 2007 | [6] | M | X | X | X | X | X | X | | US |
| Shuey & Willson, 2008 | [7] | S | X | X | X | X | X | | | US |
| Yang & Lee, 2009 | [8] | P, F, S, M | X | X | X | | X | X | | US |
| Chen, Yang, & Liu, 2010 | [9] | S | X | X | X | X | | | | C |
| Yang & Lee, 2010 | [10] | P | X | X | X | | X | | | US |
| Sacker, Worth, & McDonough, 2011 | [32] | S | X | X | | X | | X | | DK GER UK US |
| Warner & Brown, 2011 | [11] | F | X | X | | | X | X | X | US |
| Brown, O'Rand, & Adkins, 2012 | [12] | F, P | X | X | X | | X | X | X | US |
| Hu et al., 2015 | [35] | F | X | X | | | | | | CEE[1] |
| Marshall et al., 2015 | [13] | P | X | X | X | | | X | | UK |
| Xu et al., 2015 | [14] | P, M, C | X | X | X | X | | | | US |
| Boen, 2016 | [16] | S | X | X | X | X | X | | X | US |
| Ferraro, Schafer, & Wilkinson, 2016 | [17] | P | X | X | X | X | | | X | US |
| Leopold, 2016 | [18] | S | X | X | X | X | | X | | SE |
| Leopold 2018 | [33] | P | X | X | X | X | X | X | | US UK NLSE |
| Leopold & Leopold 2018 | [19] | S | X | X | X | X | | X | | GER |

*(Continued)*

**Table 1.** (Continued)

| Author(s), Year | | Health dimension | Development | Cohort | Panel attrition | SES | Race | Sex | Early Life | Country |
|---|---|---|---|---|---|---|---|---|---|---|
| Leopold 2019 | [20] | S, P | X | X | X | X | X | X | | GER |
| Lu, Pikhart, & Sacker, 2019 | [34] | o. | X | X | | X | | X | X | US UK |
| | | | | | | | | | | C J |

Note: P: Physical health; F: Functional health; S: Self-rated health; M: Mental health; C: Cognitive health; o.: Other

[1] CEE = Central and Eastern Europe

# indicates the number of countries.

welfare policies, which change over time and are not uniformly expansive in all domains (see e.g., [47]). The UK and Germany can be regarded as key examples on this point. While firmly placed in the liberal welfare regime in terms of labour market policies, for health care, the UK is placed alongside Scandinavian countries with high levels of decommodification [48], while its rental market places it close to Germany in terms of housing policy [49]. An archetype of the Bismarckian welfare state regime based on social insurance in the 1970ies, Germany has profoundly changed its welfare policies in the last two decades in a sequential reform trajectory, towards a hybrid, dual welfare system that embraces liberal aspects of welfare systems for labour market "outsiders" in combination with social democratic provisions for the "insiders" [50, 51]. Our choice to compare England and Germany therefore is not rooted in a choice of contrasting cases in the framework of essentialist and reductive welfare typologies, but rather from an analytical interest: How does the association of education and physical health limitations, and its development over time, vary, depending on three crucial institutional aspects of inequality: (1) how education is distributed, (2) how education is associated with health inequalities and (3) the accessibility of health care systems.

First, one of the most important features of the twentieth century was the increase in both the average level and duration of schooling in more recently born cohorts. Importantly, educational expansion happened in an idiosyncratic way in each country. Both in England and Germany, access was widened to secondary education (after the Second World War) and to higher education (in the late sixties) in a similar timeframe [52], but resulted in stark differences in completed levels of education for each cohort, as starting positions differed for both countries: As Table 2 shows, in 1950, a higher proportion of the population had completed upper secondary education in Germany compared to England 20 years later. However, England always has had a slightly higher proportion of the population completing post-secondary education. For the cohorts under study, in England, the majority of the population had lower levels of education than in Germany. The division of Germany from the Second World War until 1989 led to slightly different paces of expansion within Germany. For the former Eastern part of Germany, the expansion was slightly slower and higher education was slightly less accessible [53].

Second, differences in educational levels are associated with health disparities in later life [60], but the extent of these differences vary considerably by country [61]. While a plethora of theoretical perspectives discusses the existence and extent of different health inequalities in modern welfare states (for an overview see [62]), we focus on the role of education as a vector for life chances in a given context. While education has some absolute influences on health through the knowledge acquired, from a life course perspective most of its effects are relative and context dependent. Education can be considered a positional good, which means it is more valuable if less of it is around, at least when it comes to its effect on subjective well-being [63] or on occupational access [64]. This paper wants to explore if education also functions as

**Table 2. Key macro-level indicators on education and health (care) in England and Germany.**

| Aspect | Source | Indicator | England | Germany |
|---|---|---|---|---|
| Educational attainment | [54] | Upper Secondary (1950), in % of population | 11 | 20 |
| | | Upper Secondary (1960) | 13 | 24 |
| | | Upper Secondary (1970) | 16 | 28 |
| | | Post Secondary (1950) | 7 | 6 |
| | | Post Secondary (1960) | 8 | 7 |
| | | Post Secondary (1970) | 10 | 9 |
| Education to health pathway | [55] | Activity limitation, 50–64 (based on EU SILC), in % of age group | | |
| | | Low educated | 36 | 51 |
| | | Middle educated | 25 | 41 |
| | | High educated | 17 | 28 |
| | | Activity limitation, 65–79 (based on EU SILC), in % of age group | | |
| | | Low educated | 43 | 61 |
| | | Middle educated | 34 | 57 |
| | | High educated | 28 | 53 |
| | [56] | Education-related health inequality, over age 50 (Gini-coefficient) | .02 | .01 |
| Health care | [57] | Current healthcare expenditure relative to GDP (2017), in % | 9.6 | 11.3 |
| | [58] | Health insurance coverage, in % | 100 | 89.4 |
| | [59] | Number of physicians per 1,000 of a population (2017) | 2,786 | 4,249 |
| | [58] | Life expectancy (2018), in years | 81.4 | 81.3 |

a positional good in terms of later life health. Previous studies found that the association of educational inequalities on functional health in later life is smaller in Germany than in England, especially in older cohorts [56], even if the absolute level of functional health limitations is higher in Germany [55]. This study wants to place these findings in a longitudinal context, which will allow us to make an analytic distinction between age and cohort level influences.

Third, differences in healthcare systems can explain differences in how consequential health conditions are and how well they can be managed. England and Germany differ substantially in the way healthcare is organised and delivered. While in England health care is state funded and free at point of use, in Germany it is organised through a compulsory public health insurance system, which is deducted from wages. Health care spending is higher in Germany, which has more hospital beds, a higher proportion of doctors, and more generally more health infrastructure than the UK. Nevertheless both provide universal access and comprehensive coverage to their respective populations [65].

## CAD in health in three key tenets

This paper highlights the comparative aspect of CAD. To do so, it also follows the three central tenets of CAD outlined by Wilson, Shuey and Elder [3], which we summarize below.

**a) Life course approaches.** A life course approach emphasizes the age-graded patterns of human development over the lifespan. Cumulative (dis)advantage theory (CAD) emphasizes how differences are patterned systemically, by initial level of advantage. One common way of classifying (dis)advantages over the life course is by using educational level as a marker of social position. Education in this vein is often seen as a vector of life chances, providing access to better paid employment, with fewer health risks, enabling more comfortable housing in safer neighbourhoods. A lower level of education in this way is linked to more stressful living

and working environments, as well as more risky health behaviours. Accordingly, research has shown a link between less education and additional psychosocial risk factors for poor health, such as the earlier onset of poverty, unemployment or separation, as well as having fewer resources for coping with them [66].

As such, CAD is the central paradigm of studies from a life course perspective connecting socio-economic position, either measured over different points of the life course [9, 10, 15, 21, 28, 37, 67] or seen as educational level [18, 19, 25, 33], and the cumulative burden of health insults it entails. These health insults depend on the health dimension under consideration. Previous research on CAD processes using a comparative perspective focussed on subjective health [32] or physical health [33]. In the current study we focus on functional health, as it is less affected by psychological re-adjustment processes than subjective health (i.e., how ill people feel), and it is less related to differences in health care systems than physical health (i.e., which diagnosed illnesses people have) [68]. Nevertheless, functional health limitations are a strong predictor of both individual outcomes such as subjective well-being and societal consequences such as health and social service costs [70].

**b) Separating age and cohort effects.** A second feature of CAD is to emphasize that effects of age and cohort should be disentangled [69–71]. As presented in Table 1, previous research did not always follow this central tenet of the CAD paradigm: Both in epidemiological research, with a strong focus on population averages, and in social science research, with a tradition of group specific health levels, a mix of cross-sectional, longitudinal, and longitudinal cohort-sensitive research designs was used. While longitudinal approaches tackle the individual process of ageing, cohort changes reflect historical developments in the circumstances under which people grow up and grow old. Disentangling these two processes, as done by an increasing number of longitudinal cohort-sensitive research projects, including the current study, helps to disentangle the effect of both.

Health does not change in a constant rhythm but may alter its speed. Each health dimension follows a particular development over age. While subjective health for instance has been shown to decline slightly over age, functional health development is high in early life stages and decreases–with accelerated speed–in later ages [8, 12, 25]. Accordingly, non-linearity should be taken into account. In order to investigate how CAD in health unfolds as people age, it is important to account for cohort effects [72, 73]. Different birth cohorts experience varying life conditions, so that levels of accumulated risk exposure differ between them. Earlier cohorts, born before or during the Second World War, generally have grown up under worse circumstances. Experiencing an economic crisis at the beginning of their working life for example has been shown to exert a scarring influence on later life health [74]. Cohort effects in this way capture the impact of historical conditions on a person's life. It is widely assumed that later-born cohorts are healthier than earlier-born cohorts, reflecting a multidimensional process of improvement in medical care, reduction of environmental pollution, less physically demanding work and better lifestyle habits [75]. Importantly, these cohort effects often differ by social status groups. Willson and colleagues [3] showed that higher levels of education in later-born cohorts lead to better self-rated health relative to earlier-born cohorts. For those with lower levels of education, however, self-rated health depends on the life stage considered, with better health earlier in life, but worse health later in life relative to earlier-born cohorts. Importantly, failing to separate age from cohort effects implicitly assumes that rates of change in health over age are the same across all cohorts [69, 73, 76]. In sum, there are strong reasons to assume that birth cohorts affect how CAD processes unfold.

**c) Selective panel dropout.** When investigating health changes over age, addressing potential attrition is crucial [3, 72]. Over time, drop out is potentially higher among those with declining health, so that the slope of the average population trajectory might be

underestimated, or positively biased [72]. Specifically, dropout is higher at lower levels of education, as they have worse health already at the beginning of the observed period. This selective participation of the lower educated may lead to an overestimation of the average health status in this group. It has been shown that the decline in the health gradient at older ages (i.e., age as a leveller of social differences) relies partly on this type of selective attrition [3, 73]. While selective participation might obscure a part of the picture, real mortality changes the scenery entirely. Because the least healthy part of a disadvantaged population segment such as the lower educated, die earlier, average health levels in this group increase [77]. Thus, tackling panel attrition in studies of CAD is not only a matter of method but also a matter of substance.

### The current study

We assume that the mean level of health at the age of 50 is on a similar level in England and in Germany (H: *Country*). However, as education affects health stronger in England than in Germany, we believe there will be greater health differences between educational groups at baseline in England than in Germany (*H*: *Country*\**Education*). Based on CAD, or the idea that differences at the beginning lead to larger (dis)advantages later in life, we expect the social gradient to increase over age (including its acceleration) stronger in England than in Germany (*H*: *Country*\**Development*\**Education*). In terms of education, we believe a more unequal distribution leads to starker health disparities. As such, we assume that education has smaller effects for more recent cohorts, and more so in Germany than in England (*H*: *Country*\**Cohort*\**Education).*

Summing up, we compare functional health development in later life in Germany and England, two countries with similar health care accessibility and historical patterns of educational expansion but a differing distribution of education. By comparing health developments in England and Germany, we investigate if differences between educational groups and their change over time depend on the social context. Accordingly, we put the CAD processes into context. Table 3 sums up the hypotheses.

## Methods

### Data

Our analyses use the English Longitudinal Study of Ageing (ELSA; [78]) and the German subsample of the Survey of Health, Ageing and Retirement in Europe (SHARE; [79]). ELSA is a panel study of the community-residing English population age 50 and over (at baseline) and

**Table 3. Summary of the hypotheses.**

| Label | Hypothesis |
|---|---|
| *Country* | Average limitations in ADL are the same in England and Germany. |
| *Country*\**Development* | With increasing age, limitations in ADL increase at the same rate in England and Germany. |
| *Country*\**Education* | There is a greater difference in limitations in ADL between people with higher and lower education in England than in Germany. |
| *Country*\**Development*\**Education* | The educational gradient in limitations in ADL increases with age in England to a greater extent than in Germany. |
| *Country*\**Cohort*\**Education* | The influence of education on limitations in ADL does not change differently by country over cohorts. |

**Table 4. Mean levels, standard deviations, and number of observations for ADL by wave, educational level, and country (means and standard deviations are weighted cross-sectionally).**

| | | Wave | | | | | |
|---|---|---|---|---|---|---|---|
| | | 1 | 2 | 4 | 5 | 6 | |
| Interview year* | | 2004 | 2006 | 2010 | 2012 | 2014 | Total |
| Lower educated | | | | | | | |
| England | Mean | 0.52 | 0.52 | 0.53 | 0.50 | 0.47 | 0.51 |
| | SD | 1.04 | 1.04 | 1.04 | 1.03 | 0.99 | 1.03 |
| | N | 3,631 | 3,280 | 2,505 | 2,369 | 1,974 | 13,759 |
| Germany | Mean | 0.29 | 0.42 | 0.51 | 0.44 | 0.40 | 0.39 |
| | SD | 0.75 | 1.04 | 0.96 | 1.01 | 1.02 | 0.94 |
| | N | 338 | 260 | 112 | 484 | 305 | 1,499 |
| Middle educated | | | | | | | |
| England | Mean | 0.31 | 0.27 | 0.29 | 0.28 | 0.26 | 0.28 |
| | SD | 0.82 | 0.78 | 0.79 | 0.80 | 0.80 | 0.80 |
| | N | 3,748 | 3,951 | 3,344 | 3,526 | 3,253 | 17,822 |
| Germany | Mean | 0.19 | 0.18 | 0.24 | 0.21 | 0.20 | 0.20 |
| | SD | 0.67 | 0.67 | 0.78 | 0.68 | 0.65 | 0.68 |
| | N | 1,102 | 934 | 511 | 2,145 | 1,544 | 6,236 |
| Higher educated | | | | | | | |
| England | Mean | 0.20 | 0.15 | 0.17 | 0.19 | 0.14 | 0.17 |
| | SD | 0.73 | 0.59 | 0.65 | 0.72 | 0.59 | 0.66 |
| | N | 1,103 | 1,321 | 1,182 | 1,177 | 1,067 | 5,850 |
| Germany | Mean | 0.08 | 0.11 | 0.13 | 0.09 | 0.10 | 0.10 |
| | SD | 0.43 | 0.55 | 0.55 | 0.44 | 0.49 | 0.47 |
| | N | 524 | 453 | 282 | 1,073 | 828 | 3,160 |
| Overall | | | | | | | |
| England | Mean | 0.39 | 0.36 | 0.37 | 0.35 | 0.32 | 0.36 |
| | SD | 0.93 | 0.89 | 0.89 | 0.89 | 0.85 | 0.89 |
| | N | 8,482 | 8,552 | 7,031 | 7,072 | 6,294 | 37,431 |
| Germany | Mean | 0.18 | 0.21 | 0.24 | 0.21 | 0.19 | 0.20 |
| | SD | 0.64 | 0.73 | 0.76 | 0.69 | 0.67 | 0.68 |
| | N | 1,964 | 1,647 | 905 | 3,702 | 2,677 | 10,895 |
| Total | N | 10,446 | 10,199 | 7,936 | 10,774 | 8,971 | 48,326 |

* Both data sets collect data around turn of the year, accordingly interview year indicates the year of beginning data collection.

their partners, conducted biannually since 2002. Ethical approval for all the ELSA waves was granted from NHS Research Ethics Committees under the National Research and Ethics Service (NRES). SHARE is a cross-national panel study conducted every two years since 2004. Ethical approval for SHARE was granted during Waves 1 to 4 by the Ethics Committee of the University of Mannheim and for the following waves by the Ethics Council of the Max Planck Society. For the current study, no further ethical approval is required. We used the harmonized version of the data, available through the Gateway to Global Ageing project. We restricted the age range from 50 to 90 as there are limited respondents over 90 (ELSA: 0.8%; SHARE: 2.3%). We only used waves common between both studies (2004 to 2014, excluding 2008, as the health items were not part of SHARE that wave, see Table 4). As our focus is population comparisons, we only need randomly assigned respondents and hence excluded partner data from both studies (ELSA: 19.8%; SHARE: 35.8%) but retain all available information for the main

respondents, including proxy interviews (ELSA: 3.2%; SHARE: 2.8%). Refreshment samples were included (ELSA: 14.1%, in w3, w6, w7; SHARE: 58.5%, in w2 and w4) to reduce selective panel dropout. The refreshment sample of ELSA wave 4 was omitted as information on education was not comparable. Educational information was complete in SHARE, but 7.9% was missing in ELSA. Based on studies showing that partners tend to share similar levels of education [80], if possible, we substituted missing information on education with the partner's education level (5.4%). This approach reduces systematic drop-out although it might lead to minor misclassifications: For instance, women stronger profited from the educational expansions which in turn means that in particular men of earlier-born cohorts might receive lower levels of education in case of missing information [80]. For 2.5% of the sample, no partner information was available thus we omitted these individuals from the analyses. We excluded all participants with missing data on any of the study items (ELSA: 0.3%; SHARE: 0.2%). This results in a sample of 11,352 individuals with 37,431 observations for England and 5,573 individuals with 10,895 observations for Germany (see Table 4). Note that the mean number of observations differs between England ($n_i$ = 4.0, SD = 1.3) and Germany ($n_i$ = 2.5, SD = 1.3), less because of major differences in retention rates but rather due to the relatively larger refreshment samples included in the later waves of SHARE Germany [81, 82].

## Dependent variable: Health-related limitations in activities of daily living

Our dependent variable is limitations in activities of daily living due to health reasons (ADL). ADL represents the functional dimension of health (i.e., how health limits people activities) which strongly impacts well-being. Respondents indicated whether they had any difficulty (yes/no) doing five self-care tasks (difficulties with bathing or showering; dressing, including shoes and socks; eating, cutting up food; getting in or out of bed; walking across a room) which we summed up to a count variable ranging from 0 (no limitations) to 5 (limitations in all tasks). While further limitations have been assessed in both studies, the item selection is in line with other important panel studies, for instance, the US Health and Retirement Study (HRS) and the Japanese Study on Aging and Retirement (JSTAR).

## Independent variables

Based on the International Standard Classification of Education (ISCED), we distinguished between people with a low level of education (reference category; up to lower secondary), middle level of education (higher secondary), and high level of education (tertiary and more). Cohorts were categorized into 6 groups based on the year of birth. The number of cohorts is a trade-off between sufficient case numbers within each cohort and the aim to allow for a detailed analysis of differences between birth cohorts. Except for the first and the last cohort, which capture 9 and 10 years respectively, cohorts cover an 8-year period (1914–22 = 0; 1923–30 = 1; 1931–38 = 2; 1939–46 = 3; 1947–54 = 4; 1955–64 = 5). Age was recoded starting with youngest age 50 and was divided by 20 for layout purposes. Since functional health limitations differ by gender [8, 19], in all models we either statistically controlled for gender or estimated models separately for men and women.

We additionally controlled for whether the participant was part of a refreshment sample (ELSA: 19.8%; SHARE: 58.5%) and whether the data was from a proxy interview (ELSA: 3.5%; SHARE: 2.4%). To control for structural differences still present in the two formerly divided parts of Germany, we include a variable indicating the current residency (0 "Western part", 1 "Eastern part"). All other control variables were mean-centred for each country. To check the robustness of our findings, we tested whether controlling for patterns of study participation affected the results. We coded study participation as either complete (reference category;

ELSA: 56.2%; SHARE: 59.6%), selective (participation in at least two waves, but not complete; ELSA: 43.6%; SHARE: 21.6%), or one-time only (ELSA: 1.0%; SHARE: 18.8%).

## Analytical strategy

We used longitudinal Poisson population-averaged regression models for count data in stata14 (xtgee, family(poisson)) to estimate how ADL changed as age increased [83]. These models are very similar to linear growth curve models (with random intercepts and fixed slopes), except that an increase of one unit in the dependent variable is not linearly associated with an increasing independent variable but may vary over its range. Age and cohort effects are disentangled by including interaction terms between age and cohort, to allow analysis of the shape of 12-year trajectories of ADL by birth cohort. The 12-year age range is based on a 11 years of observation period of the applied data (2004–2014) plus one year due to our categorisation of age groups. As shown in S1 Table in S1 File, a person born in 1948 is 56 years old at the first observation in 2004. He or she can be traced until age 66 in 2014. Due to the clustering of age in 2-years intervals, a person born one year later would be included in the same health trajectory (age 57 to 67). Visually, the cohort-specific health trajectories are represented through age vector plots (for an introduction to this approach, see [73, 76]; for empirical examples see e.g., [9, 13, 84]). The entire stata code necessary to replicate the analyses is publicly available at OSF (see Data Availability).

In the first model, we separately assessed cohort-specific trajectories of ADL over age in England and Germany (Models 1a and 1b). To examine the educational stratification of health, as well as how health inequalities evolved with increasing age, in the second model we added educational level, and its interaction terms with age and cohort (Models 2a and 2b). To test whether countries differ significantly, data from both countries were pooled and an interaction effect with country was included for all parameters (Model 3). We used England as the reference category which means that these effects represent how Germany differs in these parameters to those estimated for England. For the opposite direction, a model using reverse coding comparing England to Germany can be found in the Supplement (S2 Table in S1 File). To test for gender differences, we estimated the country-comparison model for men and women separately (Model 4a and 4b).

As ignoring selective panel attrition can result in underestimating health inequalities with age, we estimated two additional models to examine to what extent this is the case (Model 5a and 5b). The first model excluded indicators for respondents' participation patterns (complete, selective, one-time) and being part of the refreshment sample to identify the extent of increasing selection bias [8, 9]. Subsequently, we estimated the same model with full-information maximum likelihood (FIML). FIML uses the covariance matrix to estimate each participant's ADL at any point, regardless of whether the respondent actually participated in the measurement occasion or not. This procedure corrects for Missing At Random (MAR) type attrition, in which missingness is predicted by the dependent and independent variables included in the model [85]. To do so, the growth model was estimated as a structural equation model in MPlus [86]. We discuss our findings in the light of robustness against panel attrition after presenting the central models. While we assume that no model allows to completely identify and control for selection and attrition bias, the robustness checks made do at least indicate the possible presence and direction of the bias.

## Results

### Separating age and cohort effects

As presented in Table 5, England and Germany show different patterns of development in the number of ADL (see Table 3, Model 1a and b). We present incidence rate ratios (IRR) for

**Table 5. Model predicting incidence rate ratios (IRR) for limitations in activities of daily living due to health reasons.**

| | Model 1a and 1b | | Model 2a and 2b | | Model 3 (pooled) | |
|---|---|---|---|---|---|---|
| | England | Germany | England | Germany | Ref.: England | Difference of Germany |
| Level at age 50 | 0.20*** | 0.07*** | 0.35*** | 0.15*** | 0.30*** | 0.60** |
| | (0.01) | (0.01) | (0.03) | (0.03) | (0.02) | (0.11) |
| Age/20 | 0.83* | 2.77*** | 0.69*** | 3.37*** | 0.72** | 4.13*** |
| | (0.07) | (0.64) | (0.07) | (1.02) | (0.07) | (1.18) |
| $(Age/20)^2$ | 2.05*** | 0.80 | 2.03*** | 0.75* | 2.08*** | 0.35*** |
| | (0.14) | (0.11) | (0.14) | (0.11) | (0.15) | (0.05) |
| Cohort | 1.22*** | 0.74*** | 1.10 | 0.54*** | 1.20*** | 0.43*** |
| | (0.06) | (0.07) | (0.06) | (0.07) | (0.06) | (0.05) |
| Cohort*Age/20 | 0.83*** | 1.29*** | 0.86*** | 1.38*** | 0.82*** | 1.82*** |
| | (0.03) | (0.08) | (0.03) | (0.12) | (0.03) | (0.16) |
| Middle education | | | 0.53*** | 0.54** | 0.52*** | 0.92 |
| (ref: low) | | | (0.05) | (0.11) | (0.05) | (0.20) |
| High education | | | 0.16*** | 0.23*** | 0.16*** | 1.32 |
| | | | (0.03) | (0.06) | (0.03) | (0.44) |
| Middle education* | | | 1.25* | 0.84 | 1.28* | 0.72 |
| Age/20 | | | (0.13) | (0.20) | (0.14) | (0.17) |
| High education* | | | 1.63* | 0.81 | 1.67* | 0.50* |
| Age/20 | | | (0.33) | (0.25) | (0.34) | (0.17) |
| Middle education* | | | 1.06 | 1.35* | 1.06 | 1.29* |
| Cohort | | | (0.05) | (0.16) | (0.05) | (0.16) |
| High education* | | | 1.37** | 1.37 | 1.36** | 1.04 |
| Cohort | | | (0.13) | (0.23) | (0.13) | (0.20) |
| Middle education* | | | 0.98 | 0.95 | 0.98 | 0.96 |
| Age/20*Cohort | | | (0.03) | (0.07) | (0.03) | (0.08) |
| High education* | | | 0.93 | 1.05 | 0.93 | 1.10 |
| Age/20#Cohort | | | (0.06) | (0.11) | (0.06) | (0.13) |
| N of individuals | 11352 | 5573 | 11352 | 5573 | 16925 | |
| N of observations | 37431 | 10895 | 37431 | 10895 | 48326 | |
| $\chi^2$ | 4258.86 | 1985.16 | 4367.18 | 2000.58 | 6410.43 | |
| Degrees of freedom | 9 | 10 | 17 | 18 | 31 | |

* p<0.05

** p<0.01

*** p<0.001; Displayed are incidence ratio rates (IRR) with standard errors in parentheses. Cohort is coded from earlier-born to more recent-born cohorts. Models 1 + 2 are estimated for each country independently. Model 3 uses pooled data of both countries and estimates differences in each parameter (for Germany to England) by using an interaction of a country dummy with each parameter (e.g., ADL = f(level, country, age, age*country, age², age²*country, ...)). All models are also controlled for being part of the refreshment sample, interview was conducted by a proxy person, participation pattern, gender and east or west Germany (see S2 Table in S1 File).

Poisson models, which indicate a rate ratio in the dependent count variable if the independent variable increases by one unit while the other variables are held constant. For example, an IRR of 1 in one independent variable indicates no change in ADLs, while 0.95 points to a 5 percent decline. In England, ADLs increase over age in a non-linear fashion (Age/20: $IRR_E = 0.83^*$; $(Age/20)^2$: $IRR_E = 2.05^{***}$). Each more recent-born cohort has on average a 22 percent higher risk of having one additional ADL at a given age (coh: $IRR_E = 1.22^{***}$) while showing a slower development over age (coh*age/20: $IRR_E = 0.83^{***}$). In Germany, ADLs increase linearly over

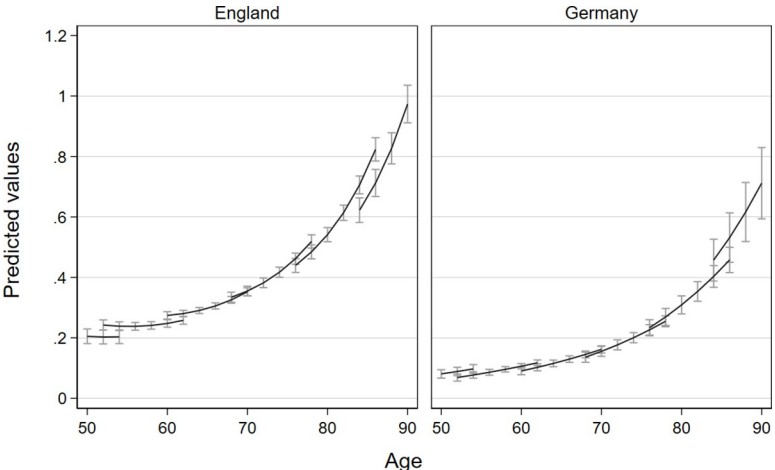

**Fig 1. Comparison of trajectories of limitations in activities of daily living due to health reasons with confidence intervals for different birth cohorts in England and Germany.** Vector-age-cohort plots of Models 1a and 1b based on separate poisson growth curve models for England and Germany, separate lines represent the succession of six cohorts.

age with strong risk increases of 177 percent points every 20 years (Age/20: $IRR_G = 2.77^{***}$). With each more recent-born cohort, in Germany, the risk of having an (additional) ADL decreases by 26 percent (coh: $IRR_G = 0.74^{***}$) while the increase in ADL over age is faster for more recent-born cohorts (coh*age/20: $IRR_G = 1.29^{***}$). Fig 1 visualizes these two contrasting patterns in ADL trajectories for both countries, with ADL trajectories being driven mainly by cohort differences in England and by age in Germany. The figure also shows a lower mean level of ADLs over the whole age range for Germany compared to England.

## Educational inequality

Models 2a and 2b test whether higher levels of education lead to lower ADL and more advantageous ADL trajectories than for people with lower levels of education. Fig 2 illustrates the

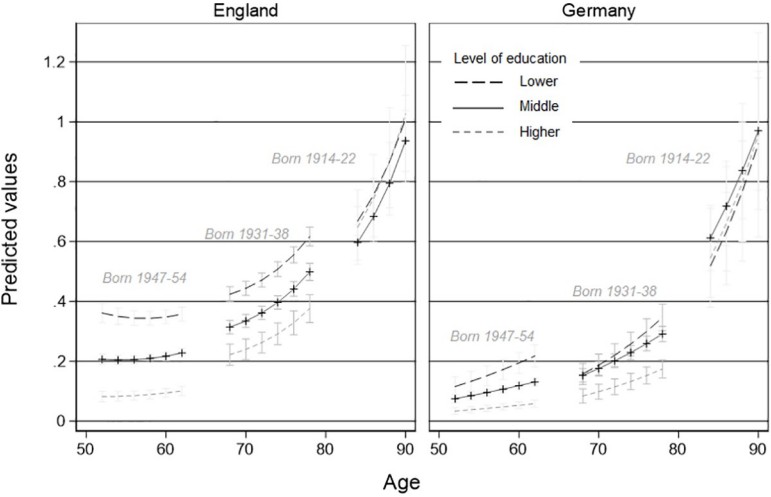

**Fig 2. Comparison of trajectories of limitations in activities of daily living due to health reasons by education and country for three exemplary cohorts.** Vector-age-cohort plot by educational level of Model 3 based on a pooled poisson-growth curve model with interaction terms of country.

results of Models 2a and 2b –for reasons of visibility–only for three exemplary cohorts (1947–54, 1931–38, 1914–22). In England, we found that, at the age of 50, the middle-educated only have half the risk of the lower educated for an (additional) ADL ($IRR_E = 0.53^{***}$) and higher educated have an even 84 percent lower risk ($IRR_E = 0.16^{***}$). In Germany, educational patterns are similar (middle education: $IRR_G = 0.54^{***}$; high education $IRR_G = 0.23^{***}$). Interestingly, because in England ADLs for middle-educated and higher-educated groups increased faster than for the lower educated (middle*age: $IRR_E = 1.25^{***}$; high*age: $IRR_E = 1.63^{***}$) educational differences in ADL decline over age, while in Germany differences between educational groups do seem to increase over age, but are not statistically significant.

However, being born in a later cohort in Germany increases the educational health gap by 35 percent for middle educated (compared to lower educated, middle*coh: $IRR_G = 1.35^*$). For the higher educated the interaction with cohort is similar in size and direction, but not statistically significant. In England, cohorts also interact with the level of limitations in ADL at a given level of education: for later-born cohorts there is a higher risk for an additional ADL for higher educated in comparison with lower educated (high*coh: $IRR_E = 1.37^{**}$). Finally, for both countries, we found no indication that age moderated the cohort by education interaction (i.e., that the improvements in health for higher educated people in more recent cohorts differ depending on age), as indicated by the not statistically significant age by cohort by education term.

## Considering country contexts

In the next step, we tested whether functional health trajectories are associated differently with education in England and Germany by pooling the data and including a country interaction term for each parameter. Table 5, Model 3 shows the coefficients for England as the reference in the first column and the difference with Germany, modelled through an interaction term, in the second column. Unexpectedly, we found that people aged 50 have a 40 percent lower risk of having an additional limitation in ADL in Germany than in England (*H*: *Country*; $IRR_{Diff} = 0.60^{**}$). Contrary to our expectation (*H*: *Country*Development*), we found that, overall, the relative risk of having an additional limitation increased 4-times faster with age in Germany than England ($IRR_{Diff} = 4.13^{***}$) but with a much slower acceleration (; $IRR_{Diff} = 0.35^{***}$). The increase in risk of an additional limitations for a laterborn cohort is smaller in Germany than in England ($IRR_{Diff} = 0.43^{***}$), but the relative risk increase by cohort over age was 82 percent greater in Germany than in England ($IRR_{Diff} = 1.82^{***}$). This indicates that later-born cohorts have a steeper increase in ADL over age in relative terms in Germany, but lower starting levels than in England.

Model 3 equally tested whether there was evidence of different patterns of CAD in England than in Germany. Although we expected to find larger health disparities between educational groups in England than in Germany (*H*: *Country*Education*), as the coefficients for the difference of Germany in terms of educational level are not significant, this hypothesis was not supported. While we expected to find larger increases in the educational gradient with age in England than in Germany (*H*: *Country*Development*Education*), Model 2 already revealed that the opposite is the case: For England, a decline of the within cohort differences between educational levels was found instead of an increase while in Germany no statistically significant trend was detected. Comparing both countries in Model 3 reveals that in Germany, higher educated tend to have smaller risk-increases over age (compared to the German lower educated) than the English higher educated ($IRR_{Diff} = 0.50^*$). Finally, there is an indication that secondary education for later-born cohorts in Germany is associated with significantly higher relative risk of an additional limitation than later-born cohorts in England ($IRR_{Diff} = 1.29^*$).

The educational gradient of ADL over age across cohorts does not differ significantly between England and Germany (*H*: *Country*∗*Education*∗*Cohort*).

## Robustness of the findings to gender differences and panel attrition

To evaluate the robustness of our findings, we estimated separate models by gender, a model without any control variables, and a model using FIML. We present these in the Supplement in full (see S2 Table in S1 File). Overall, the general pattern of results was stable, although we could identify some specific trends. First, gender differences were minor, with overall patterns being fairly similar for men and women. Second, robustness checks regarding panel attrition suggested that the overall level of limitations seems to be slightly higher, while education-based selection for level differences and their developments over age and cohort are fairly unbiased. In sum, our results appeared to represent a conservative but robust picture of health developments for different educational groups which might be even stronger in the entire population than in this sample.

## Discussion

Cumulative disadvantage, or the idea that inequalities widen between social groups over age [1, 2], has matured as a field of research into a rich and wide collection of findings, and an established set of research strategies, summarised in the three central tenets of CAD research: a life course approach, separating age from cohort effects and asserting the influence of selective panel dropout. We argue that, in line with an emerging focus on comparative CAD research, the institutional context of these processes plays an equally important role in determining if and how CAD processes emerge. The current study wants to illustrate this overlooked fact, without relying on essentialist and reductive welfare typologies, but by conducting a rigorous and in-depth comparison of the educational gradient in levels of and changes in functional health limitations as people progress from midlife to later life between England and Germany. By an analysis of limitations in ADL of in total 48.326 observations from 16.925 respondents from ELSA and the German part of SHARE in an accelerated panel design through a GEE model that treats the outcome as a Poisson distribution, as well as conducting multiple robustness checks, we can formulate an answer to our hypotheses posed earlier.

First, this study shows that the increase in limitations in activities in daily life over time is driven by different analytical forces in both countries. While in England cohort differences explain the majority of variation over time in ADL, in Germany it is more due to age evolution. This underlines the importance of the basic tenets of CAD in a life course tradition: To study development of health over age it is important to differentiate age and cohort effects. We illustrate that the mean-level ADL development over time is driven by a specific mixture of age and cohort effects in each country. While in England more recent-born cohorts showed on worse health at the age of 50 they also had fewer steep increases over age than earlier-born cohorts. For this mean-level effect exactly the opposite is true in Germany: more recent-born cohorts have better health for a given age, but steeper relative changes with increasing age. While the distinction between age and cohort effects might seem analytical and academic, to some extent it reflects the difference between onset of ill health due to biological and societal reasons. From this perspective the cohort driven deterioration of functional health trajectories in England is counter intuitive, but has been shown in similar studies looking at mental health, wellbeing [84] and frailty [13].

Second, we found that in England educational differences in ADL decline over age, while in Germany differences between educational groups do seem to increase over age, but are not statistically significant. So, we did not find a process of CAD in the strict sense of diverging health

trajectories. On the contrary, we find processes of convergence over age, in particular for England, in which lower educational levels develop "less disadvantageous" than the other groups leading to more similar health levels in later life. A partial explanation of this fact could lie in a higher risk of early mortality, leaving "healthy survivors" in the population [77]. In particular in the group of the lower educated, this might lead to increases in mean levels of health and accordingly to smaller differences to the other educational classes. While this contradicts most research on CAD [3, 13, 16, 87], it aligns with findings from Switzerland [5] and it tends in the same direction of an earlier examination of England. The latter showed stable educational gaps for physical health but was limited to a smaller age and cohort range [33]. Earlier research that found increasing educational gradients over age focussed on more subjective markers of health such as self-rated health [18] or used a fixed effects approach which obscures cohort and country differences [25]. The current study hence contributes to an understanding that CAD is not a universal pattern in Western societies [5]. Rather it shows that using an analytical approach to compare CAD processes in two similar countries is useful: It questions the generalizability of the CAD assumptions and helps to better understand differences in CAD processes by referring to institutional particularities, such as the distribution of education and its influence on health developments over age on one hand, and differing age cohort associations with health on the other hand.

More in general, two parallel processes reducing social differences seem to take place: first a life cycle effect, with increasing ADLs over age (*Education\*Development*), and second a cohort effect, pointing out that in earlier-born cohorts educational differences are less pronounced than in later-born cohorts (*Education\*Cohort*). While in England both processes occur simultaneously and independently (no effect of *Education\*Development\*Cohort*), in Germany health differences between educational classes increase mainly due to cohort succession.

These findings can be interpreted in the light of institutional differences between England and Germany highlighted earlier, namely (1) accessibility of health care, (2) how education is distributed, and (3) how education is associated with health inequalities. Regarding the first, although England and Germany follow different traditions of health care organisation, the differences found should not be based on accessibility because both countries provide full access free of charge. Second, in England, the cohorts under study have, on average, lower levels of education than those in Germany. As lower education is associated with lower functional health, this compositional effect might explain the higher mean level of ADLs in England at the age of 50. A similar speed of educational expansion in both countries (see Table 2 and [52]) might affect the succession of education-based life chances over cohort similarly, however, due to the different initial level between the two countries, in England in particular the higher educated in later-born cohorts show stronger increases in functional limitations while in Germany this effect can be found for both middle and higher educated. Third, previous studies found that education related inequalities on functional health in later life are smaller in Germany than in England, especially in earlier-born cohorts [39]. The current study points out that CAD in England follows a function of age and cohort while CAD in Germany follow a function of cohort only, leading to similar levels of functional health inequality in the intersection of higher ages and earlier-born cohorts for England and Germany.

Nevertheless, the current examination of trajectories and educational gradients in functional health across the second half of life has specific socio-political implications. The functional dimension of health is more strongly related to the need for care, rehabilitation and medical assistance than, for instance, indicators of physical or emotional health [68]. Our findings hence suggest that–in particular in England but also Germany–the lower educated need to spend more financial resources for aid with daily life activities over major parts of the life course in both countries. This is alarming as people with lower levels of education already have

fewer economic resources but must invest more in order to maintain quality of life in the face of higher functional limitations. Over the life cycle differences decline, resulting in a need for support independently of education. Thus, health care systems in both England and Germany should be aware for the early needs for support of lower social status groups. Tackling the educational gradients at age 50 suggest that welfare state investments earlier in life may reduce cumulative costs and hence may lead to lower overall costs by reducing health expenditure in later life.

## Strengths, limitations and further research

The study shows several strengths. Strongly guided by the life course approach, the present study examined educational inequalities in functional health by separating age from cohort effects in two particular contexts. Specifically, we examined the extent to which functional limitations in later life depended on a person's level of education. Acknowledging differences between health dimensions, we focussed on functional health. Recently, a growing amount of research focussed on CAD processes in different country contexts. For our comparison, we applied an analytical approach to identify potential drivers of CAD in health (care) institutions. Our results show that both average health development as well as CAD processes unfold in country-specific ways. We provided particular attention to selective panel attrition, which is of utmost importance in research on health and ageing.

Some limitations have to be mentioned. Due to the main focus of the paper on the interaction of health development, educational class, and country comparison, we acknowledged cohort-effects only as a linear trend. However, cohort changes in health might be curvilinear or cohort-specific in response to critical historical events. In a similar vein, period effects addressing for instance economic changes are out of scope of the current paper. Secondly, we combined two data sets with different case numbers to examine the overall development of functional limitations. Instead of weakening the model by reducing the observations of England to the number in Germany, we used all observations available. Our approach aims to keep maximum accuracy of the estimates. However, smaller case numbers in the German data set might lead to lower chances to find significant differences to England. It appears that Germany tends to have better functional health levels and smaller educational differences at the age of 50 when models are estimated separately, but since in the pooled dataset these differences tend to lack statistical significance, we assume that with upcoming data points in future differences between data sets might become estimated more precisely and hence the context effects might become even more apparent. Third, it is possible that the educational level in the English context does not adequately mirror the function it has in Germany, as a vector for life chances. It has been pointed out that occupational position in contrast to educational attainment has more salience in the England [28, 88] as a framework for studying health inequality. Due to different occupational coding in both surveys, this route could not be explored.

Based on our finding that Germany and England show different "rhythms of ageing" (in terms development of limitations), with German individuals remaining on a lower level of limitations for longer, and accelerating stronger in later life than the English, future research should focus on potential reasons behind this. While we also show that these rhythms differ by cohorts, institutional differences for particular cohorts might help to inform which effects lead to the different cohort developments [89] and which institutional arrangements also affect the intra-cohort inequalities. Better understanding in each country context might contribute to a better understanding of CAD as within-cohort divergence. Furthermore, future work should take into account that these processes might depend on the health domain and on the life stage. For instance, previous studies [18, 27] have shown that CAD in health is not a particular

later-life process but might appear already in earlier life phases. Since health is a complex, multi-dimensional construct [68], researchers need to be very clear about which specific dimension of health they analyse. Using a stronger systematic approach with accounting for health dimension, cohort and context differences, previous ambivalent findings primarily from the US [3, 10, 16] could become interpreted more clearly.

## Conclusion

CAD in functional health is a process which can be better understood by comparing different contexts. The current study indicates that CAD processes cannot be detected in England and Germany after the age of 50, as differences between educational groups decline by age in England and do not increase significantly in Germany. We found, however, increases in health differences between educational groups for later-born cohorts in both countries, which can be understood as increases in inequality between cohorts, due to educational expansion. If the Brechtian worker lived in current times, his poetic statement on the role of social position for health status, would benefit from referring to age, cohort and country, to be able to be assessed in terms of its plausibility.

## Supporting information

**S1 File.**
(DOCX)

## Acknowledgments

This paper uses data from SHARE Waves 1, 2, 4, 5, and 6 (DOIs: 10.6103/SHARE.w1.611, 10.6103/SHARE.w2.611, 10.6103/SHARE.w4.611, 10.6103/SHARE.w5.611, 10.6103/SHARE.w6.611, see [79] for methodological details) and from ELSA Waves 1 to 7, see [78]). We thank everybody who contributed to these projects. Finally, this analysis uses data and information from the Harmonized ELSA (Version E as of April 2017) and SHARE (Version D.4 as of March 2018) dataset and codebook developed by the Gateway to Global Aging Data. For more information, please refer to www.g2aging.org. We thank the Gateway to Global Aging project who reduced the data management tremendously.

## Author Contributions

**Conceptualization:** Martin Wetzel, Bram Vanhoutte.

**Data curation:** Martin Wetzel.

**Formal analysis:** Martin Wetzel.

**Investigation:** Martin Wetzel, Bram Vanhoutte.

**Writing – original draft:** Martin Wetzel.

**Writing – review & editing:** Martin Wetzel, Bram Vanhoutte.

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
