## [Decision Letter · Decision Letter 0]

27 Feb 2020

PONE-D-20-01985

Putting cumulative (dis)advantages in context: Later-life health trajectories in England and Germany.

PLOS ONE

Dear Dr Wetzel,

Thank you for submitting your manuscript to PLOS ONE. After careful consideration, we feel that it has merit but does not fully meet PLOS ONE’s publication criteria as it currently stands. Therefore, we invite you to submit a revised version of the manuscript that addresses the points raised during the review process.

The reviewers raise a number of methodological queries and have requested further contextual information to develop the justification for the cross-country comparison. Please address each of the reviewers' comments carefully, either by making changes to the manuscript, or providing a clear explanation in the response to reviewers for why you have not made the changes suggested. I would like to add a couple of minor points to those raised by the reviewers. One is to be careful not to conflate relatively low levels of educational attainment with inequalities in educational attainment. At the bottom of page 8, you base some of your hypotheses on greater educational inequality in England compared with Germany. However, in the background section you show that England had much lower levels of attainment than Germany, on average, in the 1970s; this doesn't necessarily equate to greater inequality. If England did indeed have greater levels of inequality, please provide that evidence in the background section. Also, you cite two studies that have previously investigated educational inequalities in functioning in these two countries (Jurges 2009 and Cambois et al 2009). Please make explicit in the text how your work builds and improves on this previous evidence. 

We would appreciate receiving your revised manuscript by 31 May 2020. To enhance the reproducibility of your results, we recommend that if applicable you deposit your laboratory protocols in protocols.io, where a protocol can be assigned its own identifier (DOI) such that it can be cited independently in the future. For instructions see: http://journals.plos.org/plosone/s/submission-guidelines#loc-laboratory-protocols

We look forward to receiving your revised manuscript.

Kind regards,

Anne McMunn

Academic Editor

PLOS ONE

Journal Requirements:

Reviewers' comments:

Reviewer's Responses to Questions

**Comments to the Author**

1. Is the manuscript technically sound, and do the data support the conclusions?

Reviewer #1: Partly

Reviewer #2: Partly

Reviewer #3: Yes

2. Has the statistical analysis been performed appropriately and rigorously? 

Reviewer #1: I Don't Know

Reviewer #2: N/A

Reviewer #3: Yes

3. Have the authors made all data underlying the findings in their manuscript fully available?

Reviewer #1: Yes

Reviewer #2: Yes

Reviewer #3: Yes

4. Is the manuscript presented in an intelligible fashion and written in standard English?

Reviewer #1: Yes

Reviewer #2: Yes

Reviewer #3: Yes

5. Review Comments to the Author

Reviewer #1: The paper has an interesting premise, but I find that there are some major conceptual issues that the authors should address.

1. A major question I would like the authors to address is whether we can we really study cohort effects with these data, and whether the analyses really disentangle age and cohort, as the overlap between age and cohorts is very limited. I think some of the results reported stem from extrapolating a lot across time, e.g. we do not know what ADLs were like at age 50 for the earlier-born cohorts, as they have not been observed at this age. Period effects also remain unaddressed.

2. The modelling assumes that there is a linear trend in the outcome by successive cohorts, for which there is no strong justification in the text. Different cohorts may have differences between them because of factors related to e.g. specific conditions such as war and famine that the cohorts experienced in a sensitive age. The authors have not tried to check to what extent a linear trend across cohorts would be the best way to conceptualise cohort differences by comparing the cohorts by categorising this variable. I would recommend that the authors justify how they treat the cohort variable based on both theoretical grounds as well as by checking the data.

3. Why did the authors include age and age2 in the models? I didn’t see any justification for this or whether other polynomials were tested.

4. Why do the interactions with age only interact with age and not age2? It is very difficult to interpret the interactions because they seem to ignore the non-linearity in the age effects that has been added in the models. I am not sure if the interactions have therefore been interpreted correctly. I think the conclusions should be drawn from what the figures show, rather than the statistical significance of interaction terms. For example, on page 25 line 3 it is stated that there is a higher acceleration in ADLs in Germany than England, but it doesn’t look that way from Figure 1.

5. Could the fact that there an individual can have a maximum of 5 ADLs create a ceiling effect that may also explain any potential reduction in educational differences by age? E.g. by age the number of ADLs increase in all educational groups, but because this variable is restricted to a maximum 5, it means that the highest educated would catch up with the lowest educated in later age.

Minor issues:

6. p.14 line 1-2 Is the wording here correct? “increase in one unit in the dependent variable” or “independent” variable?

7. Why was participation rate in 2010 so low in the German SHARE data?

8. There are some contradictory sentences in the discussion, such as when on page 22 the authors state that prior CAD research hasn’t found reducing inequalities with age, but then on page 25 discusses an additional two studies (references 9 & 55) that have found that CAD is not a "later-life process". To my knowledge, it is commonly found that there is some reduction in health inequalities in later life to a large part due to survivor bias, which I have not seen the authors discussing.

9. Unclear sentences/proof reading needed: p.9 line 1-2, p.23 lines 20-21, p. 25 lines 5-7

Reviewer #2: This study aims to compare the educational gradient in functional health development among older people in England and Germany, using longitudinal data. Although this paper is interesting, the authors should strengthen the rationale and the empirical analyses before publication. Below are some questions and comments that I hope the authors find useful.

1. While the introduction reads well, as a reader I would have preferred the focus on the country context. My understanding is that the main contribution of this paper is to further the research on CAD processes in two European countries. I would therefore encourage the authors to better describe what is known about these two countries, their educational levels, and their health. For instance, you mention that in Europe the educational health gap is smaller than in the US but it would be useful to provide more details about this. Also, what have those few comparative studies found so far? And what does this study add to the existing knowledge?

2. The choice of Germany and England should be better justified. Do these countries offer sharp or nuanced contrasts with regard to the social conditions and institutional factors which may shape individual health trajectories and their associations with education? Also, given the history of Germany (with long-standing East/West divide) I would expect a better description of those policies likely to affect both the health profile and the educational gradient. Not sure whether you would have enough power, but one could also consider to analyse East and West Germany separately.

3. A better description of the countries would also help the readers understand where your hypotheses come from. For instance, I was unsure why you hypothesise that the health profile of Germany and England is similar, given that studies which have compared SHARE and ELSA tend to suggest that age-standardised limitation in one or more ADL or IADL tends to be lower in SHARE than ELSA.

4. You mention that some country and educational differences might depend on the health outcomes used. So, why did you consider only ADLs? This needs a better and more articulated justification than the one sentence on page 8 and the brief rationale on p11. Also, given that previous comparative studies have considered a range of different health measures, I wonder whether it would be worth considering other measures too to test if similar educational trajectories would hold also for chronic diseases or long-standing illnesses for instance.

5. Finally, I assume that the two countries also differ in the extent of educational disparities in smoking, obesity, and physical exercise (most likely with larger differences observed in England). If such cross-country educational disparities in health behaviours do exist, I wonder whether it is worth both mentioning them and also controlling for these factors in your analyses?

6. There are some methodological decisions which are currently not well described in the paper and which do not seem to be convincing

a. Why were partners excluded whereas proxy respondents were not? Did you exclude only younger partners or all partners even if they were ‘core members’ of the study? The target population of SHARE is all individuals aged 50 and older and their partners, independent of age.

b. Why did you exclude the refresher sample of ELSA in Wave 4? The IFS provide education for all sample respondents (via UK data service). Also, substituting missing values with their partner’s level of education should be better justified given gender differences in educational attainments among older people. Also, if you exclude Wave 3 for SHARE, should you not also exclude Wave 4 of ELSA to maintain comparability?

c. Germany is known to have a very high attrition rate between waves. Could you specify how many of those 5,500 respondents were interviewed at all waves? A similar description would be useful for ELSA too. At the moment, it is unclear who the respondents are and what proportion was only interviewed at only 1 wave. Also, I wonder whether it would be worth presenting Table 1 restricting the sample to those present at all waves?

d. If ELSA had refresher samples in Wave 7 (the latest one you consider), how is it possible that only 1% of your final analytical sample was a one-time off (see p.13)?

e. Why did you not consider using the toilet as one of the ADL difficulties? I might be wrong but I think both surveys include this item.

f. It appears that you have already conducted analyses separately by gender. Given that gender seems to be significant in all models and that other studies have found stronger increase in health gaps between education groups among women, I would present the models separately.

g. Can you specify how you categorise German Eastern and Western respondents? Is this based on their current residence or on where they were born?

h. I would appreciate it if you could better explain Model 3. In particular, what does the ‘difference of Germany’ column in Table 3 represent? I am not sure I understood how to interpret those values.

7. The description of you results is at times difficult to follow. I understand that you have many tables and results to show. However, it would be best to focus on those findings which highlight the main contribution of your study and compare the educational gradient in the two countries under study. The section on robustness checks could be considerably reduced.

8. Why did you present in Fig 2 only three cohorts? Commenting in the main body on just a few of the trends is fine, but it would still be useful to present a figure with all cohorts under study.

9. In your discussion, it would help readers if a clearer statement was made about what we learned from the study that we did not know before. In other words, what is the story or punch line here? The authors should make an effort at pulling the results together in an integrated fashion. Also, you should link/relate your own findings to the existing literature. How do your findings contradict previous studies? And what could be the potential reasons for such unique results for England?

Reviewer #3: I enjoyed reading this manuscript, but several issues should be addressed.

• The manuscript rightly states that “complex, multi-dimensional construct [46], researchers need to be very clear about which specific dimension of health they analyse.” Therefore, the health outcome should be explicit in the Title as well as throughout the manuscript. This study focused on ADL limitations, and we don’t know whether the results in England and Germany would differ if another outcome was used, say cognition or blood pressure. Therefore, the findings should not be overstated, especially in the Conclusion.

• The authors did a nice job with sensitivity analysis on attrition within each cohort, but it appears that the attrition was worse in German SHARE than in ELSA? Even after including the refreshment samples, the mean number of ADLs measures are 4.0 in ELSA and 2.5 in SHARE-Germany. The response rates for each study should be reported in Methods. If attrition was larger in one study than in another, this should also be mentioned as a study limitation in the Discussion. Even with careful methods to deal with attrition within a study, could this influence the differences in results found between England and Germany?

• Abstract>Background: I would replace the word “development” with “decline,” since technically both baseline samples include people who have already developed functional limitations.

• The Background>Country context should be strengthened with a more critical review of the available evidence. Findings from previous studies are not very informative because the health outcomes in these studies are only reported for a few of them. If possible, more attention should be paid on the studies analysing objective health outcomes instead of self-rated health which can be problematic since the concept can change as people age due to changing expectations, as well as between cultures. If word count is an issue, the earlier sections can be cut down.

• Is it really true that that CAD processes have generally been examined in single countries?

http://dx.doi.org/10.1136/jech-2015-206548

https://doi.org/10.1017/S0144686X19001740

The first paper looks at CAD processes in physical functioning in 3 European countries.

The 2nd paper doesn’t mention the hypothesis explicitly but shows differences in CAD processes between 4 countries.

These are only two examples, but available evidence should be reviewed appropriately and included in the paper as relevant. Depending on the available evidence, the authors should reconsider the claim that cross-country research in this area is largely absent.

• Methods>Page 10: It’s the English Longitudinal Study of Ageing, not on Ageing

• Table 2: This should report medians and IQR since ADLs are a count. If ADL scores ranging from 0-5 and the means are all below 1, this shows the data are not normally distributed.

• Results>page 18: The ELSA results IRR 0.12 and IRR 1.23 don’t match the IRRs reported in the Table. The pvalue for IRR 1.63 is below 0.05 in the Table but below 0.001 in the text. Please correct.

• Somewhere in the manuscript, it would be good to mention the larger public health significance for ADL limitations. Why is it important?

6. PLOS authors have the option to publish the peer review history of their article (what does this mean?). If published, this will include your full peer review and any attached files.

Reviewer #1: No

Reviewer #2: No

Reviewer #3: No

---

## [Author Response · Author response to Decision Letter 0]

27 Aug 2020

Please see the attached document.

---

## [Decision Letter · Decision Letter 1]

15 Oct 2020

PONE-D-20-01985R1

Putting cumulative (dis)advantages in context:

Comparing the role of educational inequality in later-life functional health trajectories in England and Germany.

PLOS ONE

Dear Dr. Wetzel,

Thank you for submitting your manuscript to PLOS ONE. After careful consideration, we feel that it has merit but does not fully meet PLOS ONE’s publication criteria as it currently stands. Therefore, we invite you to submit a revised version of the manuscript that addresses the points raised during the review process.

We look forward to receiving your revised manuscript.

Kind regards,

Anne McMunn

Academic Editor

PLOS ONE

Additional Editor Comments (if provided):

Dear Martin,

You and your colleagues have done an excellent job of responding to review comments. Reviewer 1 continues to have some queries regarding the manuscript, some of which would be useful to address before publication.

R1 asks for an explanation regarding the sentence 'The 12-years age range is based on 10 years of observation plus 2 additional years one at the beginning and one at the end of the period which rely n the clustering of the age variable in 2-year brackets" on p. 18. I am equally unclear as to what you mean by this. Please could you explain?

R1 also mentions formatting changes to Fig 1 and adding a legend to Fig 2 which I agree would help the readers' understanding of these figures.

In their original comments on the finding of reducing inequality in older ages, R1 raised the issue of survivor bias. You have not commented on this in your reply and have not added it to your Discussion text as a potential explanation for these age patterns. Please could you add this or explain why you feel it is not appropriate to add this?

Finally, In your response to the first of the 'Minor issues' raised by R1 you include some new text explaining the poisson GEE models but it looks to me like the old text is still included in the analytic strategy explanation on p. 18? R1 also notes a word missing in line 18 on this page and R2 suggests a careful edit which I agree would be helpful.

These are only minor edits/queries but I think they will improve the manuscript for publication once they are done.

Yours sincerely,

Anne

Reviewers' comments:

Reviewer's Responses to Questions

**Comments to the Author**

1. If the authors have adequately addressed your comments raised in a previous round of review and you feel that this manuscript is now acceptable for publication, you may indicate that here to bypass the “Comments to the Author” section, enter your conflict of interest statement in the “Confidential to Editor” section, and submit your "Accept" recommendation.

Reviewer #1: (No Response)

Reviewer #2: All comments have been addressed

Reviewer #3: All comments have been addressed

2. Is the manuscript technically sound, and do the data support the conclusions?

Reviewer #1: Partly

Reviewer #2: Yes

Reviewer #3: Yes

3. Has the statistical analysis been performed appropriately and rigorously? 

Reviewer #1: I Don't Know

Reviewer #2: Yes

Reviewer #3: Yes

4. Have the authors made all data underlying the findings in their manuscript fully available?

Reviewer #1: Yes

Reviewer #2: Yes

Reviewer #3: Yes

5. Is the manuscript presented in an intelligible fashion and written in standard English?

Reviewer #1: Yes

Reviewer #2: Yes

Reviewer #3: Yes

6. Review Comments to the Author

Reviewer #1: Unfortunately, I still struggle to follow the analysis in this paper and to understand whether the conclusions are supported by the data, as the inferences seem to derive from the multiple interaction terms, which are difficult to follow. I am unable to comment on the suitability of the statistical models, but here are a few suggestions to improve the reader’s experience:

- Please could you be clearer by what you mean by “The 12-years age range is based on 10 years of observation plus 2 additional years one at the beginning and one at the end of the period which rely on the clustering of the age variable in 2-years bracket” (page 18 line 8-10)

-Please could you correct “To test whether countries differ statistical significance,” (page 18, line 18)?

- It would help if the authors summarized the key results at very start of the discussion.

- Figure 1: currently the panels for England and Germany are of different widths. It would help the comparison if they were the same.

- Figure 2: could a legend be incorporated into the figure, rather than being explained in the note below the figure? Also it would be help if you could see in the figure region which cohorts the lines pertain to.

- Could you explain what you mean by confirmatory bias in page 25 line 14? Why is survivor bias not discussed here?

- The wording “as differences between educational groups decline both by age and cohort in England, and in Germany educational differences decline only over cohort” in the conclusion is confusing, as educational differences increase in more recent cohorts. It is worth being more precise in language throughout the manuscript so that the message is clear.

Reviewer #2: I think that the authors have done a good job responding to the reviewers' comments and have written a very interesting paper. However, although I myself am not a native-English speaker, I encourage the authors to pay attention to errors in the English language (and to odd sentence construction) and to proof-read the article carefully. See, for instance, in the abstract 'more and more'; why is there an acronym if you only mention ADLs once?; should you perhaps add trajectories "of ADLs" among 6 birth cohorts? Or "depending" on p12.

Reviewer #3: The authors have addressed my peer review comments, plus other updates to strengthen the manuscript. Great work!

7. PLOS authors have the option to publish the peer review history of their article (what does this mean?). If published, this will include your full peer review and any attached files.

Reviewer #1: No

Reviewer #2: No

Reviewer #3: **Yes: **Milagros A Ruiz

---

## [Author Response · Author response to Decision Letter 1]

27 Nov 2020

Please refer to the word document.

---

## [Editor Report · Decision Letter 2]

9 Dec 2020

Putting cumulative (dis)advantages in context:

Comparing the role of educational inequality in later-life functional health trajectories in England and Germany.

PONE-D-20-01985R2

Dear Dr. Wetzel,

We’re pleased to inform you that your manuscript has been judged scientifically suitable for publication and will be formally accepted for publication once it meets all outstanding technical requirements.

Kind regards,

Anne McMunn

Academic Editor

PLOS ONE

Additional Editor Comments (optional):

Thank you for making these final revisions. I'm happy to accept this for publication.
---

## [Editor Report · Acceptance letter]

16 Dec 2020

PONE-D-20-01985R2 

Putting cumulative (dis)advantages in context:
Comparing the role of educational inequality in later-life functional health trajectories in England and Germany. 

Dear Dr. Wetzel:

I'm pleased to inform you that your manuscript has been deemed suitable for publication in PLOS ONE. Congratulations! Your manuscript is now with our production department. 

Kind regards, 

on behalf of

Prof Anne McMunn 

Academic Editor

PLOS ONE